# Characterization of Mutational Tolerance of a Viral RNA–Protein Interaction

**DOI:** 10.3390/v11050479

**Published:** 2019-05-25

**Authors:** Maria A. Prostova, Elena Smertina, Denis V. Bakhmutov, Anna A. Gasparyan, Elena V. Khitrina, Marina S. Kolesnikova, Anna A. Shishova, Anatoly P. Gmyl, Vadim I. Agol

**Affiliations:** 1Institute of Poliomyelitis, M. P. Chumakov Center for Research and Development of Immunobiological Products, Russian Academy of Sciences, 108819 Moscow, Russia; prostovna@gmail.com (M.A.P.); l-smertina@mail.ru (E.S.); labagola@mail.ru (D.V.B.); g.anna07@mail.ru (A.A.G.); khitrina62@mail.ru (E.V.K.); bakhmutov@mail.ru (M.S.K.); a_shishova@list.ru (A.A.S.); apgmyl@mail.ru (A.P.G.); 2Institute of Molecular Genetics, Russian Academy of Sciences, 123182 Moscow, Russia; 3Faculty of Fundamental Medicine, M. V. Lomonosov Moscow State University, 117192 Moscow, Russia; 4Faculty of Biology, M. V. Lomonosov Moscow State University, 119991 Moscow, Russia; 5A. N. Belozersky Institute of Physical-Chemical Biology, M. V. Lomonosov Moscow State University, 119899 Moscow, Russia; 6Sechenov First Moscow State Medical University, 119991 Moscow, Russia

**Keywords:** poliovirus, RNA, *cis*-elements, RNA/protein interaction, protease 3C, nucleotide/amino acid sequences, randomization, SELEX, mutational robustness, viability

## Abstract

Replication of RNA viruses is generally markedly error-prone. Nevertheless, these viruses usually retain their identity under more or less constant conditions due to different mechanisms of mutation tolerance. However, there exists only limited information on quantitative aspects of the mutational tolerance of distinct viral functions. To address this problem, we used here as a model the interaction between a replicative *cis*-acting RNA element (*ori*L) of poliovirus and its ligand (viral protein 3CD). The mutational tolerance of a conserved tripeptide of 3CD, directly involved in this interaction, was investigated. Randomization of the relevant codons and reverse genetics were used to define the space of viability-compatible sequences. Surprisingly, at least 11 different amino acid substitutions in this tripeptide were not lethal. Several altered viruses exhibited wild-type-like phenotypes, whereas debilitated (but viable) genomes could increase their fitness by the acquisition of reversions or compensatory mutations. Together with our study on the tolerance of *ori*L (Prostova et al., 2015), the results demonstrate that at least 42 out of 51 possible nucleotide replacements within the two relevant genomic regions are viability-compatible. These results provide new insights into structural aspects of an important viral function as well as into the general problems of viral mutational robustness and evolution.

## 1. Introduction

Genome replication of RNA-containing viruses is generally characterized by a high level of mutations due to the infidelity of their RNA-dependent RNA polymerases and the overwhelming absence of proofreading. The error rate varies in the range of ~10^−3^–10^−6^ nt/site/copying act [1,2,3]. Thus, newly generated molecules of poliovirus RNA (~7.4 × 10^3^ nt) differ from their templates averagely at one nucleotide position [4]. Consequently, even clonal populations of such viruses exist as highly heterogeneous swarms of variants referred to as quasispecies [5,6]. Nevertheless, under constant conditions, RNA viruses demonstrate a marked conservation of their genetic structure owing mainly to the selective disadvantage of debilitated mutants. However, viral intra- and inter-host transmissions often involve bottlenecks, upon which haphazardly picked-up genomes establish new lineages. Although such situations may contribute to the viral evolvability, they are also associated, especially if occurring repeatedly, with high risks of severe fitness cost or even the dead-end of lineages, a phenomenon known as the Muller ratchet [7,8]. To counteract or minimize effects of such disadvantageous developments, viruses possess mechanisms ensuring their relative genetic robustness, which include such well-known phenomena as degeneracy of the genetic code or of spatial structures of functional elements of RNA or protein molecules [9]. However, there exists only limited information concerning detailed and quantitative aspects of the mutational tolerance of distinct viral functions.

Interactions between elements of viral RNA and proteins required for efficient viral reproduction are attractive objects with which to study this problem. Quantitative evaluation of functional effects of mutational alterations of these partners can provide deeper insights into not only the nature of robustness of these interactions but also of their mechanistic details. As a model, we used here the poliovirus, a member of the *Enterovirus* genus of the Picornaviridae family of animal viruses. Its positive-strand RNA genome harbors a single open reading frame (ORF) encoding the polyprotein, which is eventually processed into a dozen mature proteins [10,11]. Among others, this set includes the RNA-dependent RNA polymerase 3D^pol^, the viral primer protein VPg (also known as 3B), and the protease 3C^pro^ responsible for the polyprotein processing. The ORF is flanked by the 5′- and 3′-untranslated regions, containing, in particular, two *cis*-elements required for efficient RNA replication: *ori*L, also known as the cloverleaf because of its secondary structure [12,13], and *ori*R, respectively (Figure 1).

In particular, *ori*L is involved in initiation of the synthesis of both the viral (positive) and complementary (negative) RNA strands [13,14,15,16]. An essential step in accomplishing these functions is by recruiting *ori*L of several viral and host proteins to generate a ribonucleoprotein (RNP) complex [13,17,18]. A key reaction in the formation of the relevant RNP is the interaction of the hairpin-like domain *d* of *ori*L with the viral protein 3CD, the uncleaved precursor of 3C^pro^ and 3D^pol^ [17,18,19,20] (Figure 1). Specifically, the apical tetraloop of domain *d* is believed to interact with the TGK motif of the 3C moiety of 3CD, since alterations in the structure of the former may be functionally compensated by the mutational changes in the latter [20,21,22].

In a previous study [20] we assayed the level of mutational robustness of the domain *d* of *ori*L by the randomization (in the context of the full genome) of the sequence of its tetraloop and two adjoining base pairs, followed by selection and investigation of the viable progeny (the SELEX procedure). In addition to such in vitro selection experiments, certain altered genomes were engineered and investigated. Unexpectedly, it was found that the overwhelming majority, if not all, of the possible point mutations of this highly conserved octanucleotide were compatible with the viral viability. Some of these mutations did not change the viral phenotype, whereas others inflicted functional defects of variable strength, as evidenced by the impairment of viral reproduction, RNA synthesis, and *ori*L/3CD binding, these three alterations well correlating with each other. The debilitated mutants demonstrated a good potential for phenotypic recovery by either acquisition of advantageous mutations in the octanucleotide or by changes in the TGK motif of the 3CD protein, and in particular its conversion into IGK or TGR.

To further characterize the mutational tolerance of the poliovirus *ori*L/3CD interaction, the TGK-encoding nonanucleotide of 3C was now subjected to randomization. After deriving some preliminary conclusions from the selection and sequencing of viable viruses, an additional set of mutant genomes was engineered and investigated. At least 11 amino acid replacements in TGK were found to be compatible with the virus viability. The results markedly enlarged the space of 3CD sequences compatible with the virus viability and in combination with our previous study on mutational tolerance of *ori*L [20], they demonstrated that in the two genome segments controlling the tetraloop/TGK interaction (encompassing the apical octanucleotide of domain *d* of *ori*L and the TGK-encoding nonanucleotide, together 17 nt-long), at least 42 out of 51 theoretically possible nucleotide replacements are not lethal. These results not only provide further insights into the mechanistic aspects of an important step of enterovirus reproduction but also give a deeper understanding of the mutational tolerance and evolvability of RNA viruses.

## 2. Materials and Methods

### 2.1. Generation of Plasmids Encoding Poliovirus Genomes with Randomized Segments

Plasmids with the randomized nucleotides encoding amino acids 154–156 of the viral 3C protein were prepared as follows (Figure 2). A 68 nt-long synthetic DNA 3CTGK (Table 1) corresponding to the region with coordinates 5875–5942 of the poliovirus RNA but with the randomized relevant nonanucleotide (positions 5897–5905) and containing a synonymous marker mutation G_5914_C, was purchased from Syntol (Moscow Russia).

For its PCR-amplification, the reaction mixture containing 0.6 pmole 3CTGK and 20 pmoles each of 3CMluI and 3CSacII primers (Table 1) was subjected to 10 cycles of heating/cooling (20 s at 55 °C, 20 s at 72 °C, 15 s at 95 °C) and the product was treated with *Sac*II and *Mlu*I endonucleases (Fermentas, Vilnius, Lithuania) and purified by electrophoresis in 2.5% agarose. The DNA fragment thus generated was inserted into poliovirus genome-harboring plasmids in a two-step procedure aimed to minimize possible contamination with non-mutated genomes. First, it was inserted at the proper position into the plasmid pT7PV1RibMS, which contained the full-genome copy of the type 1 poliovirus RNA with the artificially created unique cleavage sites for the restriction endonucleases *Mlu*I (at position 5861/5862) and *Sac*II (5952/5953). To this end, the large fragment obtained after treatment of pT7PV1RibMS with *Mlu*I and *Sac*II was purified by electrophoresis in 1% agarose and ligated with the above-described fragment containing the randomized nonanucleotide. The created plasmid was used for the transformation of *E. coli* TOP10 (Invitrogen, Carlsbad, CA, USA) and the relevant region of a clone (pT7PV1RibMSa, clone #18 in Table 3) was sequenced and found to lack G5906, resulting in the frameshifting of the open reading frame downstream of the randomized nonanucleotide. This intermediate plasmid was used as a donor of the large SacII/MluI fragment in the ligation reaction with the small SacII/MluI fragment harboring the randomized codons, as described above. Pools of the randomized plasmids were obtained upon transformation of *E. coli* TOP10 cells.

To generate plasmids with randomized octanucleotide of *ori*L and nonanucleotide of the 3C gene, clone #18 (pT7PV1RibMSa) was treated with *Apa*I and *Spl*I and the larger ApaI/SplI fragment was ligated with the small ApaI/SplI fragment obtained by PCR with the previously described [20] synthetic DNA possessing randomized relevant octanucleotide in *ori*L. The plasmid thus generated was propagated in *E. coli* and pooled ~2250 plasmids (pT7PV1RibMSb) were further treated as pT7PV1RibMSa in the protocol for the generation of plasmids with the randomized 9 nt in 3C.

### 2.2. Construction of Genomes with Desired Mutations

Since the viruses generated by the SELEX procedure not infrequently contained mutations outside the randomized nonanucleotide, certain analogous genomes lacking these undesired mutations were constructed. To this end, RNA from the preparations of the relevant viruses generated from the randomized plasmids was reverse transcribed and PCR-amplified using 3CMluI and 3CSacII primers. The product was digested with *Mlu*I and *Sac*II and used to generate constructs encoding the full-length viral genome, as described above.

To generate plasmids encoding viral genomes with novel mutations, two separate PCR amplifications were performed using pT7PV1RibMS as the template and pairs of primers, mut-s and 3EP4, in one reaction, and mut-a and B5594 in the other, where mut-s and mut-a harbored the desired mutation (Table 1). The products obtained in these reactions were fused together in an additional PCR with 3EP17 and B5594 primers, digested with *Sac*II and *Mlu*I, and used to generate plasmids encoding the full-length viral genome as above.

For the experiments requiring purification of mutated 3CD proteins, the small MluI-SacII fragment with the desired mutation was generated by treatment with *Mlu*I and *Sac*II of the PCR product of the reaction containing pT7PV1RibMS/mut as the template and 3CMluI and 3CSacII as primers. The gel-purified fragments were ligated to the large fragment obtained upon digestion with the same endonucleases of the expression plasmid pQE60-3CD [20], which harbored the 3CD sequence of wild-type poliovirus with the H_40_A mutation (inactivating the proteolytic activity of the protein) and the N-terminal His-tag.

The presence of introduced mutations in all the engineered plasmid clones was verified by sequencing.

### 2.3. E. coli Transformation and Preparation of DNA

*E. coli* TOP10 cells were transformed by the plasmids and the plasmid DNA was isolated from individual or pooled clones as described previously [20].

### 2.4. Transcription and Transfection

The procedures were performed as described in [20]. Briefly, DNA samples from 19 plasmid pools containing from 31 to ~2250 variants were linearized by digestion with *Eco*RI and transcribed by T7pol (Thermo Scientific, Carlsbad, CA, USA), and the transcription mixtures were used for the transfection of Vero cells (ATCC® CCL-81™) as described [23]. The concentration of RNA in the transcripts was evaluated by EtBr staining in 1% agarose gels. For the experiments with engineered genomes, the viral RNA was purified by centrifugation in 5–20% sucrose gradients and the concentration of RNA was determined spectrophotometrically.

### 2.5. Sequencing of the Viral Genomes

The material from a plaque or virus suspension was suspended in 0.3 mL of the nutrient medium. RNA was isolated using either a phenol-chloroform extraction, Trizol reagent (Invitrogen), or Qiagen RNAeasy kit (Qiagen, Hilden, Germany), and was reverse transcribed using random or 3EP4 primers (Table 1). The *ori*L-containing PCR products were prepared with primers Rib2 and DEN3, whereas primers B5594 and 3EP17 were used for the preparation of DNA fragments encoding the portions of 3C that included the relevant tripeptide. The PCR products were gel-purified and sequenced either manually using afmol^®^ DNA Cycle Sequencing System (Promega, Fitchburg, WI, USA) or by automatic sequencers Beckman Coulter Seq 8000 or ABI 3130 Genetic Analyzer.

### 2.6. Time-Course of Viral RNA Replication

The experiments were performed as described in [20] with minor modifications. Vero cells monolayers grown in 12-well panels (Corning Incorporated, Corning, NY, USA), ~2.4 × 10^5^ cells/well, were transfected with 50 ng RNA transcripts per well, and the total RNA was extracted with Trizol reagent at 0, 12, 16, 20, and 24 h post-transfection (p.t.). Three wells were used as parallels for each time point. One μg of the purified RNA was used for reverse transcription with a random hexamer primer and the Maxima Reverse Transcriptase (Thermo Fisher Scientific, Carlsbad, CA USA). The standard curve was generated by serial dilutions of the wild-type transcript supplemented with 1 μg RNA from mock-transfected Vero cells. Real-time PCR was carried out using a ABI 7500 Real Time PCR System analyzer with primers PVL1 and PVR1 and FAM-tagged oligonucleotide PVP1 as the probe (Table 1).

### 2.7. Expression and Purification of Modified 3СD Proteins

The procedures used have been described previously [20]. Briefly, pQE60-3CD with the desired substitutions was used for the transformation of *E. coli* JM109 cells. The transformed cells in 200 mL of SOB medium (2% tryptone, 0.5% yeast extract, 10 mM NaCl, 2.5 mM KCl) containing 20 mM glucose and 100 μg/mL ampicillin were grown in a rotary shaker for 4–5 h at 37 °C. The medium was changed to the one lacking glucose and containing 2 mM IPTG and incubated in a rotary shaker overnight at room temperature. The cells were subjected to centrifugation and lysed by sonication, and recombinant 3CD was purified by the Ni-chelating chromatography. The electrophoretic pattern of the purified proteins is shown in Figure 3.

### 2.8. Electrophoretic Mobility Shift Assay (EMSA)

Direct interaction between the *ori*L-containing fragments of viral RNA and variants of the 3CD proteins was investigated as described in [20]. Briefly, motilities of RNA fragments corresponding to purified preparations of the 5′-terminal 115 nt of viral RNAs and 3CD proteins were determined by native 6% PAGE separately and in mixtures.

### 2.9. 3C Structure Analysis

3C structure (PDB ID 1L1N, chain A) visualization and distance labeling was performed with VMD software [24]. Electrostatic potentials at the solvent accessible surfaces of 3C were calculated using a PyMOL APBS Electrostatics Plugin with default settings [25].

## 3. Results

### 3.1. Generation of Plasmids Encoding Full-Length Poliovirus Genomes with a Randomized Tripeptide Corresponding to the TGK-Motif of the 3C Protein

As described in Materials and Methods, a set of plasmids was engineered, in which nine nucleotides (positions 5897–5905 of the poliovirus genome) encoding tripeptide TGK (positions 154–156 of the viral 3C protein) were randomized. The randomization was intentionally incomplete: the triplets in the inserted nonanucleotide contained neither A nor T at the thid positions. Although the total number of potentially encoded codons decreased from 64 to 32, their capacity to translate into all 20 amino acids was retained (the UAA and UGA stop-codons were not encoded). The decrease in the number of codons made the possible nucleotide sequence space more compact, enhancing the probability of representation of codons for all amino acids in relatively small samples of the genomes investigated.

To assess the extent of randomization, the segment corresponding to positions 5800–5960 of the poliovirus genome was sequenced in 24 randomly selected plasmids. Table 2 shows that the distribution of nucleotides across the randomized region was rather uneven, but taking into account the relatively small size of the investigated set, the level of randomization could be considered satisfactory.

Sequencing demonstrated that six plasmids did not harbor non-intended alterations in the codons of interest and adjoining regions, two plasmids exhibited some heterogeneity in the randomized sequence, and one plasmid possessed a synonymous substitution outside the randomized region (Table 3). These nine plasmids could be regarded as consistent with the goal of the experiment. One plasmid has a non-synonymous mutation outside the randomized octanucleotide, whose phenotypic effect, if any, is unknown. However, three plasmids possessed termination codons within the randomized region, and 11 plasmids had deletions, or deletions coupled with insertions, resulting in full or partial frameshifting. Thus, less than a half of the randomized plasmids could be considered as fulfilling the requirements of the experiment.

### 3.2. Selection of Viable Mutated Viruses and Primary Structures of Their Presumptive oriL-Interacting Region of 3C Proteins

Nineteen pools were assembled from different numbers (31 to ~2250) of randomly selected plasmid clones. DNA preparations isolated from these pools were transcribed in vitro and serial dilutions of these transcripts were used to transfect Vero cells. The cell cultures were observed for at least six days. All these pooled transcripts generated plaque-forming viruses, however, with variable specific infectivity, i.e., the number of plaque-forming units (pfu) per µg RNA, which was several orders of magnitude lower than that of the wild type virus (Table 4), suggesting a very low abundance of the infectious genomes in the samples. The plaques generated by different samples appeared at different times and had different sizes, suggesting the possibility that the plaque-forming capacity of some pools could be due to quasi-infectious genomes, i.e., marginally replicating RNAs, which could obtain plaque-forming capacity after the post-transfectional acquisition of additional mutations.

To characterize the variety of sequences compatible with the viral viability, a portion of the 3C-encoding region (positions 5800–5960) of the RNA from 37 randomly selected mutant clones was sequenced. In 17 such clones, the primary plaques generated upon transfection contained sufficient material for sequencing, but in other cases the material from primary plaques was subjected to one or more bulk passages. The investigated set of genomes contained 22 unique nucleotide sequences.

The prevalence of different codons in the randomized RNA region of these viruses differed markedly from that in the plasmids used for their generation (compare Table 2 and Table 5). For example, the nucleotides A and C that were underrepresented at certain positions of codons 154 and 156 of the plasmids were abundant enough in the sequenced viral RNAs. Of note is that the presence of U and A at the 3rd positions of codons 154 and 156, respectively, may indicate that the genomes of some selected viruses had indeed acquired mutations after the transfection, since the paternal plasmids should not possess the relevant nucleotides at these positions. In other words, the relevant plasmids were likely quasi-infectious. Alternatively, the mutations could be due to errors arising upon in vitro transcription.

In the selected viruses, the tripeptide at positions 154–156 of 3C was represented by 13 unique variants of amino acid sequences, including the parental TGK (Table 6). Although this set did not necessarily denote the whole variety of acceptable sequences, it revealed several important features. The tripeptide of all isolates invariably had the central Gly residue. In the genetically stable variants, position 156 of 3C could be occupied not only by Lys but also by another positively charged residue, Arg, accompanied with either the original Thr or with Val, Ile, or Cys at position 154. The latter position could also be solitarily changed into Val or Cys. Viable but genetically unstable isolates could also contain Met or Leu at position 154 (in combination with Arg_156_), which were converted upon passages into more comfortable Thr or Val. Similarly, the heterogeneity at this position in one isolate (#17) was likely due to the conversion of poor but acceptable Met in the virus that initiated infection into Val soon after the transfection. Three unstable primary isolates (with VGM, CGC, and IGW) harbored changed amino acids at both positions 154 and 156; they increased their fitness by the acquisition of a positively charged residue, Lys or Arg, at the latter position. Somewhat distinctly, a primary isolate (#21) with Ser_156_ (and Thr_154_) retained this residue upon passages but acquired Arg at the position just preceding the randomized triplet. Another virus (#22), having uncomfortable Met_156_, already possessed Arg_153_ upon isolation from the primary passage and appeared to convert, during further passages, its acceptable RTGM tetrapeptide into an even better (wild-type) CTGK.

The above results suggest some preliminary regularity. Various amino acids at position 154 are compatible with viral viability but certain ones (Thr, Val, Ile, Cis) are obviously preferable over some others (Leu and Met), the latter tending to be replaced by the former upon passaging. Position 155 appeared to be invariably occupied by Gly. Position 156 requires either Lys or Arg but tolerated to some extent non-positively charged amino acids (e.g., Met, Ser, or Trp), especially if Cys_153_ was replaced by Arg.

### 3.3. Engineering and Properties of Mutants with Definite Alterations in the 3C Protein

To define more exactly the validity of the above-formulated preliminary regularity and to obtain some information on the phenotypic properties of viruses with different 3C sequences, a number of mutant genomes were engineered and their plaque-forming capacities were compared with those obtained through the in vivo selection of the randomized RNA. This was important because the latter procedure could be accompanied by the acquisition of undetected mutations outside the randomized sequence. The results obtained with the engineered genomes are presented in Figure 4 and Table 7.

Firstly, mutants with substitutions at position 154 of 3C were engineered. In line with the results obtained with the randomized plasmids, the variants having at this position Val, Ile, or Cys exhibited phenotypes (specific infectivity as well as the time of appearance and size of plaques) comparable to those of the wild-type genome and were genetically stable. The wild-type-like phenotype was exhibited also by the virus with the TGR-encoding RNA. The replacement of Thr_154_ by Met generated a less-fit, unstable genome, which increased its fitness upon passages by either true reversion, or changing this residue to Val or Leu, or by acquiring a mutation (Pro_88_Ser) in an upstream region of the 3C sequence. However, the genome with Leu_154_ also exhibited decreased specific infectivity and plaque sizes as well as genetic instability (manifested by either Leu_154_Val replacement or the above-mentioned mutation in the nonrandomized area). The SGK-encoding genome appeared to be quasi-infectious, generating plaques at day 8 p.t. in one out of three attempts. This genome should experience two transversions after transfection to acquire a more favorable C_154_ residue, explaining the poor reproducibility of the experiments.

The replacement of Lys_156_ by Met or Ser resulted in a marked decrease in both specific infectivity and fitness, as evidenced by the very late (days 7–8 p.t.) appearance of plaque-producing viruses with either the true reversion to TGK (in the case of TGM) or a variety of compensatory mutations in other parts of 3C (in both viruses) (Figure 4B, Table 7). The set of such mutations included the appearance of positively charged Arg at position 153 just preceding the relevant tripeptide as well as various mutations in the 3C sequence upstream (Pro_88_ to Leu or Thr; Va_l101_ to lle) or downstream of it (Asp_282_ to Asn). The appearance of Arg_153_ in similar circumstances was observed in the above experiment with randomization as well. This prompted us to construct genomes with RTGM and RTGS sequences. Such viruses appeared to be markedly fitter compared to their Cys_153_-containing counterparts, as evidenced by earlier appearances of plaques, but still were genetically unstable, accumulating various mutations in other parts of 3C (Figure 4C, Table 7). These compensatory changes included mutations at already mentioned positions 88 and 101 as well as various other positions of 3С.

Notably, introduction of mutations P_88_S or I_151_L into the RTGS-containing RNA resulted in a significant gain of fitness (Figure 4C). The replacement of Lys_156_ by a negatively-charged Glu in 3C with Thr_154_ or Val_154_ generated non-infectious transcripts (Table 7).

Several genomes were reconstructed, in which alterations were introduced at positions 154 and 156 simultaneously (Figure 4A, Table 7). Some of the low-fit, genetically unstable genomes of the above set of in vivo selected viruses were reconstructed and additional variations were introduced at these positions. The MGR-containing virus exhibited a wild-type-like specific infectivity, but generated plaques later and was eventually transformed into VGR- or TGR-containing variants (similarly to the one selected from the randomized genomes). The genomes with IGW, CGC or VGM were unstable, producing late small plaques with a newly-acquired “good” Lys_156_ or Arg_156_.

It should be noted that the modifications of the TGK in the investigated cases were not accompanied with the acquisition of any alterations in its ligand, the domain *d* of *ori*L. In particular, this was demonstrated for the viruses generated by transcripts with TGM, RTGS, TGR, VGK, CGK, SGK, and IGK after a round of reproduction and viruses with MGR, MGK, IGW, LGK, and RTGM after one additional passage (not shown).

### 3.4. Checking Mutual Compatibility of Certain Altered 3СD Proteins with Altered oriL

As indicated in the introduction, previous studies demonstrated that adverse effects of certain alterations in the tetraloop of domain *d* could be, at least partially, compensated by the acquisition of some changes in the TGK motif of 3C, such as T_154_I and K_156_R [20,21,22]. Here, we investigated whether alterations of 3C could also increase fitness of the debilitated genomes with certain non-YNMG-like structures (Y = pyrimidine; N = any nucleotide; M = A or C) of the loop of domain *d*, for which such a compensatory effect was not observed earlier. The genomes were constructed with IGK- or TGR-containing 3C in combination with either agCUUGcu- or auGAGAgu-containing tetraloops of *ori*L. The former and the latter *ori*Ls appeared to confer, to the TGK-harboring genomes, low-infectious and quasi-infectious phenotypes, respectively, and both could increase their fitness by some alterations of the mutated tetraloop [20]. As shown in Figure 5**,** both IGK and TGR variants of the genomes with the non-YNMG *ori*Ls demonstrated a markedly higher fitness compared to their TGK-containing counterparts, with a somewhat lesser effect in the case of the auGAGAgu/TGR combination.

To obtain additional information about the mutual compatibility of structurally variable *ori*L/3CD ligands, plasmids were constructed that encoded full-length poliovirus genomes, in which interacting motifs of both partners (the tetraloop and two adjoining base pairs of domain *d* of *ori*L and 9 nt corresponding to codons 154–156 of the 3C gene) were randomized. Seven transcripts of pooled plasmids (~10^3^–1.5 × 10^4^ clones) were used for the transfection of Vero cells. RNA from one of the three pools with ~10^3^ plasmid clones as well as all transcripts of larger pools proved to be infectious, although with a low specific infectivity. Sequencing 113 5′-terminal nucleotides (including *ori*L) and positions 5800–5960 (a region of 3C gene) of the genome of 12 viral isolates gave the results shown in Table 8. The relevant tripeptide in 3C was represented only by the above described TGR, VGR, and LGR, the latter being replaced by the stronger VGR between the third and fifth passages. The tetraloops of domain *d* in all but one of the primary isolates possessed sequences not belonging to any consensuses with known stable structures, but they clearly tended to acquire YNMG or GNUA sequences, which have been demonstrated to be optimal for the *ori*L/3CD interaction [20]. The results indicated that VGR (similarly to the previously investigated TGR and IGK) was able to ensure certain levels of functional interaction with “bad” tetraloops, but improvements of the latter were required to ensure a more efficient *ori*L/3CD interaction. It may be noted that the unintended mutations in the *ori*L of the transcripts in pools 1 and 4 (Table 8) could hardly markedly influence the efficiency of this interaction, though the former (insertion of G at position 14) should increase the size of the loop of the hairpin *b*.

Obviously, the set of available viable viruses was too small to demonstrate the whole space of acceptable combinations.

### 3.5. Efficiency of RNA Replication of Engineered Mutants

To ascertain whether the observed alterations in the phenotypic properties caused by the mutations in the conserved tripeptide TGK were due to the changed efficiency of the genome replication, the time-course of accumulation of viral RNA in Vero cells transfected with some engineered RNAs was assayed by the quantitative PCR. Generally, the efficiency of replication correlated reasonably well with the mutant’s plaque phenotypes. The genomes with Val, Ile, and Cys at position 154 displayed almost the same replication dynamics as their wild-type Thr-containing counterpart (Figure 6A). The occurrence of Leu (Figure 6A), Met, or Ser (Figure 6B) at this position resulted in a marked decrease in the replicative capacity. Substitution of Lys_156_ by Arg was without an appreciable effect on the RNA replication (Figure 6B). Non-positively charged amino acids (Met, Ser, or Glu) at this position allowed only a marginal level of accumulation of RNA (Figure 6C). Binary mutations at positions 154 and 156 in MGR, SGR (Figure 6B) and VGE (Figure 6C) endowed the transcripts with somewhat variable but decreased efficiency. However, the replacement of wild-type Cys by Arg at position 153 of the TGM- and TGS-encoding genomes, resulting in some fitness gain, was accompanied only with a very slight, statistically insignificant enhancement of RNA accumulation (Figure 6C). Interestingly, this group with a severely damaged replicative capacity contained both low-fit but viable (possibly quasi-infectious) (TGM and TGS) and apparently dead (TGE and VGE) genomes.

Importantly, the IGK-containing mutant was able to markedly, though not fully, rescue the profoundly inhibited replication of the RNA harboring a GAGA (i.e., GNRA-type) tetraloop in *ori*L (Figure 6D), congruently with the T_154_I mutation previously observed to take place upon transfection of the GAGA-containing quasi-infectious genome [20].

### 3.6. Interaction of Mutant 3CDs with oriL

To ascertain whether phenotypic changes caused by the TGK mutations were linked to the impaired *ori*L/3CD interaction, electrophoretic mobility shift assays (EMSA) were performed. As interacting partners, we used the recombinant 3CD proteins and 5′-terminal 113 nt-long fragment of the poliovirus RNA. The results were visualized by staining with ethidium bromide (EtBr). The VGK-, IGK-, and TGR-containing proteins demonstrated a level of *ori*L-binding comparable (in the case of IGK somewhat lower than in other cases) to that of the TGK-containing control (Figure 7) in sufficiently good agreement with their phenotypes. SGK-, MGR-, and TGM-containing 3CD, conferring markedly low viral fitness, exhibited undetectable or marginal affinity to *ori*L in EMSA. The presence of positively charged Arg_153_ enhanced, though slightly, binding of the “poor” ligands. Only the CGK–containing 3CD did not show correlation between the EMSA and other phenotypes: it failed to detectably bind *ori*L but endowed the virus with acceptable fitness. The reason for this discrepancy is unknown, but it might be caused by the presence of two neighboring Cys residues (at positions 153 and 154), which could change the protein conformation under the conditions of the EMSA assay (see also the Discussion section).

## 4. Discussion

### 4.1. The Problem

The replication of RNA-containing viruses is generally error-prone, and in the case of picornaviruses nearly each act of template copying may be associated with the acquisition of a mutation [1,2,3]. Such negligence is mainly due to a low fidelity of the viral RNA-dependent RNA polymerases (RdRP) [26,27,28] and the lack (with a very few exceptions) of proofreading mechanisms. This infidelity is not an inherently incorrigible property of RdRP, since their faithfulness could be markedly enhanced by various point mutations [29,30,31,32,33]. Somewhat counterintuitively, an increase in fidelity may result in a decreased viral fitness [29,34,35,36] and hence the frequency of RdRP-made errors appears to be evolutionally tuned.

To prevent or diminish the potential harm of replicative infidelity, viruses should possess a significant degree of mutational tolerance. This tolerance is largely due to the degeneracy of codons, phenotype-neutral character of many amino acid substitutions, the ability of diverse sequences in RNA regulatory elements to maintain analogous mutual orientations, and functional equality of certain nucleotides in these elements [9]. Though the general importance of these factors for the counteracting replicative infidelity is well appreciated, only rather limited information is available on the quantitative aspects of the mutational tolerance of distinct viral functions.

### 4.2. Mutational Tolerance of the TGK Motif and the oriL/3CD Interaction

The interaction between *ori*L and 3CD is an essential step of the poliovirus genome replication [13,17,18,20,37]. This interaction involves the tetraloop of domain *d* of *ori*L and TGK tripeptide of the 3C moiety of 3CD [16,18,19,20,21,22,38]. Notably, the TGK motif is highly conserved in 3C proteins of members of the *Enterovirus* C species, though Thr, being most abundant, could also be occupied by Val, Ile, and Met [38]. This study provides insights into quantitative and mechanistic aspects of the mutational tolerance of the genome regions controlling the structure of these ligands.

As summarized in Figure 8, at least 11 nucleotide mutations out of 27 possible in the TGK-encoding nonanuclotide are compatible with the viral viability. Taken together with our previous observation of the mutational robustness of domain *d* of *ori*L [20], this means that at least 34 point mutations out of 51 possible in the two-segmented 17 nt-long stretch of RNA (octanucleotide of domain *d* and nonanucleotide of the 3C gene) are not lethal. If a second point mutation in the TGK-encoding motif is allowed (such mutations could well be already present in the quasispecies populations), then the number of the viability-compatible substitutions in it would reach at least 19 (Figure 8) and the whole space of the permitted nucleotide replacements in the 17 nt-long stretch of RNA would rise to at least 42. Additionally, the tripeptide can sustain not less than 11 amino acid replacements, these being six and five at positions 154 and 156, respectively.

Being not lethal, the amino acid replacements in 3C exerted different fitness effects. A significant proportion of them did not demonstrate, in our in vitro experiments, any marked adverse effects. Other mutations negatively affected the *ori*L/3CD interaction to different degrees, with some of them bringing the virus on the verge of a catastrophe. However, even in the most debilitating cases, the surviving viruses have a resilience tool: the infidelity of RNA replication resulting in the acquisition of reversions or compensatory mutation.

If there exist such a variety of structures of the relevant tripeptide in protein 3C with apparently more or less equal phenotypic impacts, why is TGK so strictly conserved in wild-type polioviruses? It may be speculated that the laboratory assays do not completely reflect fitness of circulating viruses. It should also be taken into account that even in tissue culture experiments competitive capacity of the relevant mutant viruses has not been assayed.

Though we are focusing here on the direct interaction between the tetraloop of domain *d* of *ori*L with the TGK motif of 3CD, it should be kept in mind that the both RNA and protein partners of this interaction have several separate functions and that the formation of the *ori*L/3CD complex involves several other viral and cellular participants and is significant not only for the RNA replication but for its translation as well [39,40,41]. The viability-compatible mutations identified in this study may affect some of these activities but obviously such effects, if any, are not virus-killing.

### 4.3. Possible Mechanistic Features of the oriL/3CD Interaction

Although the significance of the *ori*L/3CD interaction for viral RNA replication is well established, detailed information about the mechanistic aspects underlying their mutual affinity is lacking. The results reported here and in our previous paper [20], though insufficient to suggest a specific molecular model of this interaction, may nevertheless contribute to the development of such models in future.

In particular, the requirements for distinct amino acids at positions 154–156 of poliovirus 3C became partially defined. Although all full wild-type poliovirus genomes in the NCBI database have TGK in the corresponding region, the tripeptide could endure numerous modifications (Figure 8) either without any appreciable loss of fitness or with some debilitating but still viability-compatible effects. Only the central Gly_155_ appeared to be indispensable, although, admittedly, no exhaustive attempts to prove this were undertaken. The strong requirement for this residue may be related to its position at the loop between the two β-strands [42]. Gly is frequently found in loops because it provides a high flexibility to peptide chains and is often conserved as a structure determinant [43]. “Good” residues at position 154, Val and Ile, share with the wild-type Thr a methyl group at the β carbon atom, hinting that this group may be involved in a hydrophobic interaction. The sebilitating effect of Ser_154_ is in line with this assumption. On the other hand, Cys_154_, which was also able to confer a stable wild-type phenotype, has an SH group at the β carbon. It is tempting to assume that this distinction is responsible for a weak interaction of the CGK-containing 3CD with domain *d* in the EMSA assay (Figure 7). The discrepancy between this inefficiency and functional competence in the RNA replication (Figure 6) may be due to the presence of two neighboring Cys residues (see above).

Discussing the phenotypic effects of 3C mutations, additional possibilities to accomplish the *ori*L/3C interaction, e.g., via another RNA-binding motif of 3C, KFRDI at positions 82–86 [22,44], should be taken into account. Adaptive changes of Pro_88_ into Ser, Thr, or Leu observed in several viruses with unfavorable tripeptides at positions 154–156 (Table 7) may presumably be linked to the proximity of position 88 to Tyr_6_ and His_89_, involved in *ori*L recognition [45]. Pro_88_, being located in a small helix, can affect the orientation of the neighboring His_89_, which is known to interact with Tyr_6_ of 3C [42], the distance between their aromatic rings being 3.31Å (Figure 9A), which is common for stacking. These two residues have been reported also to be involved in the *ori*L/3C interaction [45] and are highly conserved in polioviruses [38]. The close proximity of His_89_ to TGK (6.43 Å and 10.66 Å to Lys_156_ and Cys_153_, respectively) and Tyr_6_ to Gly_155_ (5.28 Å) points to possible effects of substitutions in CTGK to the mutual orientation of His_89_ and Tyr_6_, which could be compensated by substitution of Pro_88_ by more conformationally flexible Ser, retaining a His_89_/Tyr_6_ interaction in the RNA-recognition.

For full functionality, position 156 could be occupied not only by the wild-type Lys but also by positively charged Arg, whereas the negatively charged Glu at this position was lethal, suggesting an electrostatic interaction in the tetraloop/3CD affinity. It may also be noted that the Lys_156_Ala replacement was reported to inhibit the capacity of 3CD to stimulate uridylylation of VPg [42], which is known to depend on the *ori*L/3CD interaction [16]. The lack of a positive charge at position 156 (in mutants with TGS and TGM tripeptides) could be partially compensated by the appearance of such a charge (e.g., in Arg) at position 153, just preceding the relevant tripeptide. Of note is that Lys153 has almost the same steric potential to interact with RNA-ligands as Lys156, as follows from the comparison of crystal structures of TGK-containing (poliovirus) and KIGQ_156_-containing (rhinovirus A2) 3C proteins: Lys_153_ of rhinovirus exposes its positive charge to the same surface area as Lys_156_ of poliovirus, though this area in the former 3C has a somewhat lower overall positive charge, due to a lesser abundance of basic amino acids [42,46] (Figure 9, compare panels (B) and (C)).

It is not clear whether debilitating effects of certain “poor” residues in the relevant tripeptide were linked to the disappearance of distinct RNA-protein interactions directly involving these residues or to changes in the protein conformation and solubility. In the latter cases, the possibility of dynamic changes of this conformation to modulate its functionality should be considered. It may be worth remembering that the functionally optimal conformation of the tetraloop of domain d of *ori*L could be provided by different sequences of the YNMG consensus, and it has been proposed that certain non–YNMG sequences are able to temporarily acquire a YNMG-like conformation as a result of molecular dynamics, acquiring thereby some level of functionality [20].

## 5. Conclusions

This study provides new quantitative and mechanistic insights into how the fundamental conflict between the infidelity of genome replication and the need to retain identity may be solved in RNA viruses. This conflict becomes especially biologically relevant in the case of various bottlenecks, often accompanying viral life history. After such bottlenecks, new viral lineages are often initiated by individual or a small number of mutated genomes haphazardly picked up from always highly heterogeneous (quasispecies) viral populations. By using as the exemplary model an essential viral RNA/protein (*ori*L/3CD) interaction required for the efficient replication of poliovirus, it was demonstrated that at least 11 amino acid replacements in the involved highly conserved key tripeptide (TGK) of the protein ligand did not kill the virus. Combining the present results with those reported by us previously [20], we concluded that at least 42 out of 51 possible nucleotide replacements within the two relevant genomic regions controlling this interaction (octa- and nonanucleotides of the *ori*L and 3C gene, respectively) could be tolerated without the loss of viral viability. A significant proportion of these acceptable mutations did not markedly affect the viral phenotype as studied in vitro. The viruses adversely affected by other viability-compatible mutations exhibited a remarkable level of resilience, i.e., the capacity to regain, fully or partially, the fitness by alterations (reversions or compensatory changes) of either RNA or protein ligands. This resilience is based again on the infidelity of viral RNA replication as well as on selection.

The levels of mutational tolerance and robustness described here could be extrapolated (with some obvious limitations related to differences in the replicative mechanisms) to other RNA viruses and thus could contribute to better understanding of their conservation and evolvability.

## Figures and Tables

**Figure 1 viruses-11-00479-f001:**
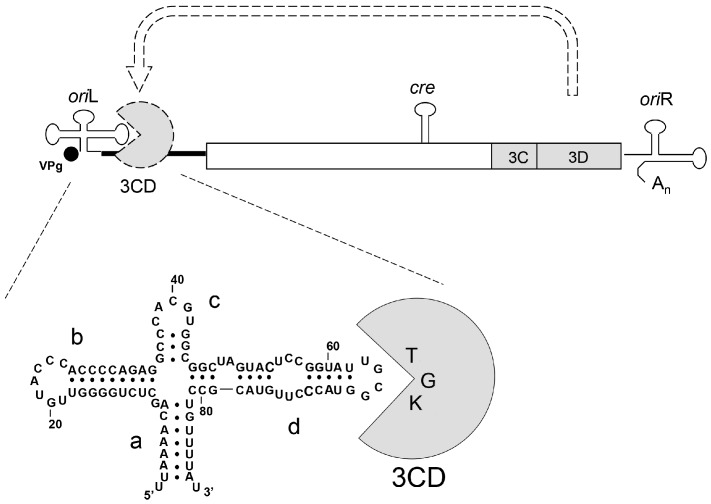
Schematic representation of the poliovirus genome, its replicative *cis*-elements, and formation of the *ori*L/3CD complex. The positions of only 3C- and 3D-coding sequences are shown in the viral polyprotein reading frame (rectangle). Locations of the major replicative *cis*-elements *(ori*L, *cre*, and *or*iR) are indicated.

**Figure 2 viruses-11-00479-f002:**
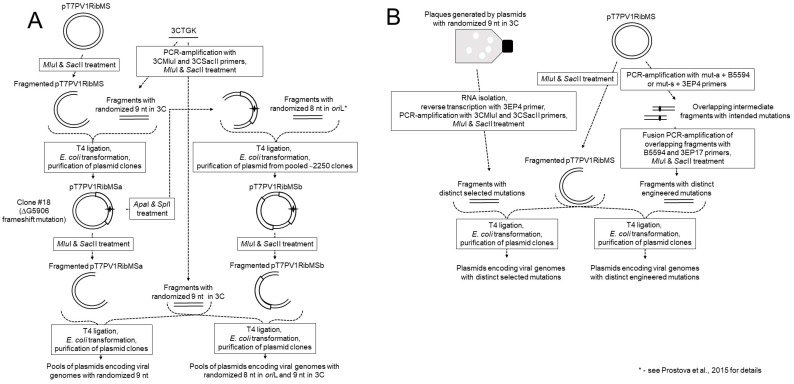
Generation of plasmids encoding viral genomes with randomized segments (**A**) or distinct mutations (**B**).

**Figure 3 viruses-11-00479-f003:**
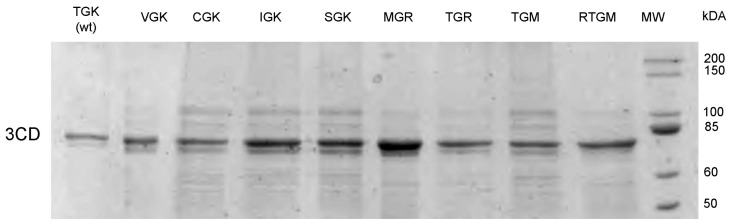
SDS PAGE profiles of the engineered mutant 3CD proteins.

**Figure 4 viruses-11-00479-f004:**
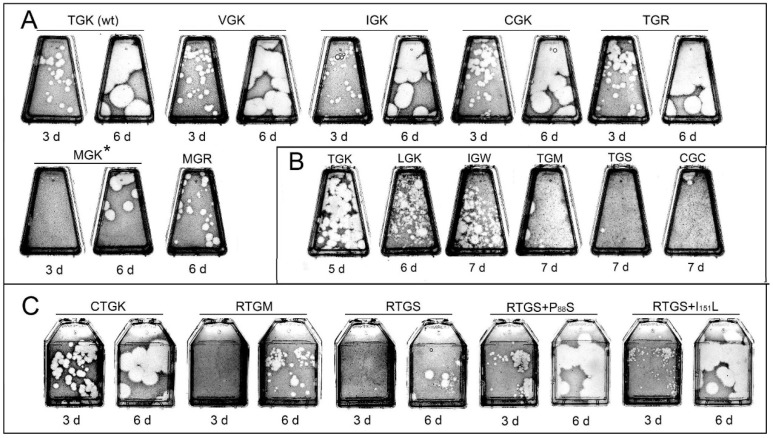
Plaque phenotypes of the engineered viruses. Fifty ng of transcripts were used for the transfection with wild type genomes in all these experiments. (**A**,**B**) Altered tripeptide TGK. Twenty to 50 ng and ~1 µg of mutant transcripts were used for the transfection of a flask in (**A**) and (**B**), respectively. (**C**) Altered tetrapeptide CTGK. Fifty to 150 ng of transcript per flask. Days post-transfection are indicated. * the same flask is shown at days 3 and 6. The results presented in panels **A**–**C** were obtained in three separate experiments, and some variability of the plaque sizes in the control (TGK-containing) samples could be due to different sensitivities of the cell cultures.

**Figure 5 viruses-11-00479-f005:**
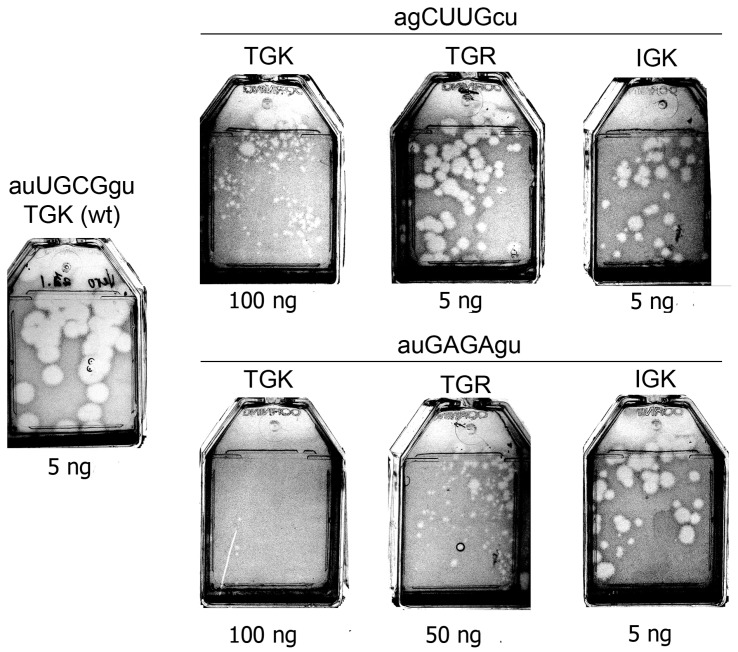
Plaque phenotypes (day 4 post transfection) of viruses in which both the apical loop of domain *d* of *ori*L and the TGK motif of 3C were modified.

**Figure 6 viruses-11-00479-f006:**
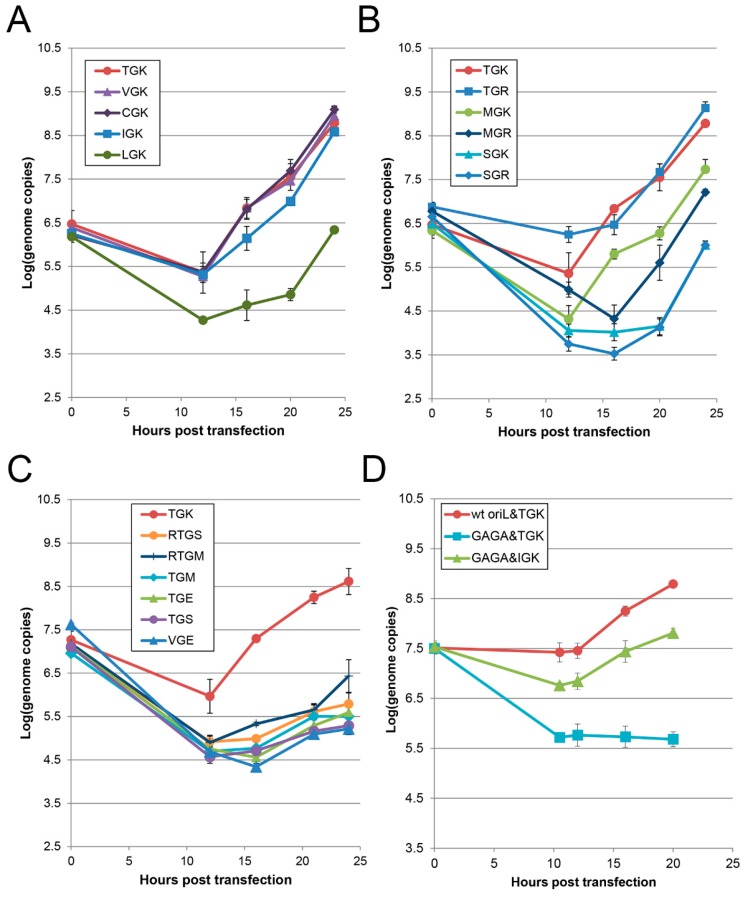
Efficiency of replication of viral genomes with modifications in only the TGK motif of 3CD (**A**–**C**) or in both *ori*L’s tetraloop and TGK (**D**). Monolayers of Vero cells were transfected with the RNAs modified as indicated and the accumulation of viral genomes was determined by quantitative PCR. Each point reflects the results obtained in triplicated samples.

**Figure 7 viruses-11-00479-f007:**
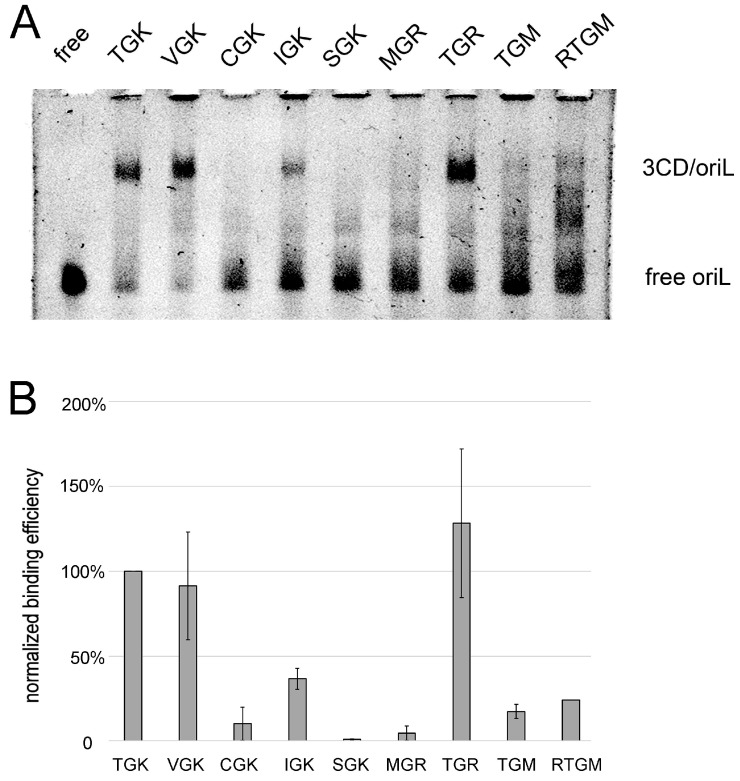
Electrophoretic mobility shift assay of the affinity between altered 3CD proteins and wild-type *ori*L. (**A**) Illustration of an experiment. The presence of 3CD in the complexes with the mobility indicated has been demonstrated previously by Western blotting [20]. (**B**) The optical densities of the *ori*L/3CD complex bands (relative to that of the TGK-containing controls) observed in two experiments (the interaction of *ori*L with the RTGM-containing 3C was investigated only once).

**Figure 8 viruses-11-00479-f008:**
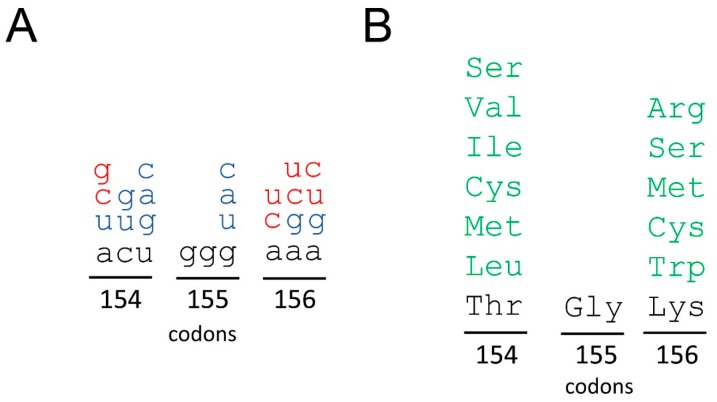
The space of nucleotide (**A**) and amino acid (**B**) sequences corresponding to codons 154–156 of the poliovirus protein 3C. The nucleotides and amino acids in the wild-type poliovirus are given in black. The set of viability-compatible nucleotide substitutions caused by single point mutation and substitutions requiring an additional mutation are given in blue and orange, respectively. The viability-compatible amino acids are in green.

**Figure 9 viruses-11-00479-f009:**
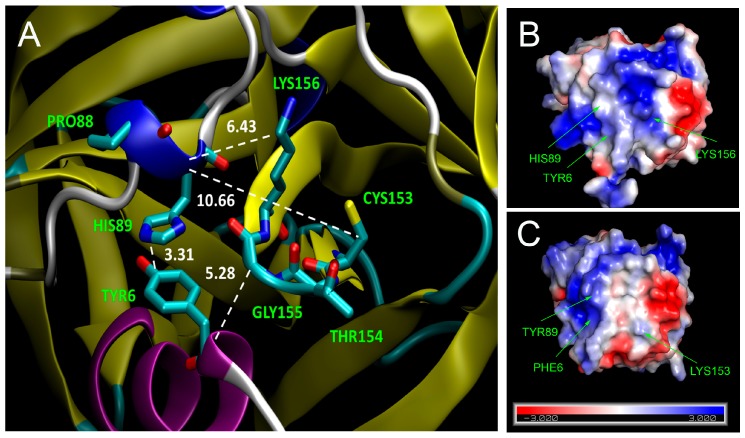
Some features of the crystal structure of 3C. Panel (**A**) (based on PDB ID 1L1N) demonstrates the proximity of the RNA-binding motif TGK to some other relevant amino acids. Oxygen, nitrogen, and carbon atoms are colored in red, blue, and cyan, respectively, and β-barrels and α-helices in yellow and blue/magenta, respectively. The distances between α carbons of His_89_, Gly_155_, Tyr_6_, β carbon of Cys_153_, ε carbon of Lys_156_, C4 of Tyr_6_, and C5 of His_89_ are indicated in Å. The electrostatic potentials of the solvent accessible surfaces of proteins 3C of poliovirus (based on PDB ID 4DCD) and rhinovirus (based on PDB ID 1CQQ), shown on panels (**B**) and (**C**), respectively, were calculated by using a PyMOL APBS Electrostatics Plugin. The potential range from negative −3 mV (blue) to positive +3 mV (red) is shown at the bottom of panel (**C**). Relevant residues (see the Discussion section) are highlighted.

**Table 1 viruses-11-00479-t001:** Oligonucleotides used.

No.	Name	Sequence ^a^	Position in the Genome
1	3CTGK	GTGTGGTGGAGTCATCACATGTnn(g/c)nn(g/c)nn(g/c)GTCATCGGcATGCATGTTGGTGGGAACGGTTCACACG	5875–5942
2	3CMluI	TTCCAACgcGtGCAGGACAGTGTGGTGGAGTCATCAC	5856–5892
3	3CSacII	CAGGGCcGCGGCAAACCCGTGTGAACCGTTCC	5928–5959, complement
4	Rib2	gaggccgaaaggccgaaaagggcctatgggcccttcTTAAAACAGCTCTGG	1–15
5	DEN3	GAAACAGAAGTGCTTGTTCG	158–177, complement
6	3EP17	CATTCTCCCATGTGAC	7082–7097, complement
7	3EP4	TTTTTTTTTTTTTTCTCCG	7436–polyA, complement
8	B5594	GAAGTGGAGATCTTGGATGCC	5594–5614
9	mut-s ^b^	CATCACATGT(mut-s)GTCATC	5892–5911
10	mut-a ^b^	(mut-a)ACATGTGATGACTCC	5902–5882, complement
11	PVP1	FAM-TTGATTCATGAATTTCCTTCATTGGCA-BHQ1	7161–7187 complement
12	PVR1	CGAACGTGATCCTGAGTGT	7212–7230, complement
13	PVL1	GGCAGACGAGAAATACCCAT	7123–7142

^a^ The mutation-generating and nonviral nucleotides are in lower case. FAM, 6-carboxyfluorescein; BHQ1, black hole quencher 1. ^b^ mut-s and mut-a stand for various distinct mutagenic nonanucleotides of sense and anti-sense polarity, respectively.

**Table 2 viruses-11-00479-t002:** The prevalence (%) of different nucleotides in the codons corresponding to amino acids 154–156 of the protein 3C in 20 randomized plasmids ^a^.

Nucleotide	Codons of the 3C-Coding Region								
	154			155			156		
	N_1_	N_2_	N_3_	N_1_	N_2_	N_3_	N_1_	N_2_	N_3_
G	25	25	70	10	45	70	60	40	90
C	20	15	30	30	35	30	5	40	10
T	45	30	0	20	10	0	20	15	0
A	10	30	0	40	10	0	15	5	0

^a^ Four plasmids (two with deletions and two with heterogeneity in the randomized region) were omitted from this analysis.

**Table 3 viruses-11-00479-t003:** The coding capacity of plasmids.

No.	Sequence of the Randomized Nonanucleotide	Encoded Tripeptide	Non-Intended Alterations ^a^
1	ggg ggg ccg	Gly Gly Pro	none
2	tgg cgg gtg	Trp Arg Val	none
3	agg acg gcg	Arg Thr Ala	none
4	ctc cgg ggc	Leu Arg Gly	t_5916_a (Met_160_Lys)
5	gcg ctg acg	Ala Leu Thr	none
6	gcc acg tcg	Ala Thr Ser	none
7	ttg ggc ttg	Leu Gly Leu	a_5884_g (synonymous)
8	tgg aac tgg	Trp Asn Trp	Δ5893, frameshift
9	cag tcg gtg	Gln Ser Val	Δ5860–5861, frameshift
10	gcg agg acg	Ala Arg Thr	Δ5893–5894, frameshift
11	ttg agc gcg	Leu Ser Ala	none
12	tag ctg gag	Stop Leu Glu	termination of translation
13	tac tсg gcg	Tyr Ser Ala	Δ5893, frameshift
14	gtc acg ggg	Val Thr Gly	Δ5895, frameshift
15	tag tcc ggg	Stop Ser Gly	termination of translation
16	agc ccc agg	Ser Pro Arg	Insertion of g at position 5882, Δ5925, local frameshift
17	cac agc tgg	His Ser Trp	Δ5895, frameshift
18	ctg cgg ggg	Leu Arg Gly	Δ5906, frameshift
19	ttg tag gcg	Leu Stop Ala	Δ5870, frameshift
20	tag agg ggc	Stop Arg Gly	termination of translation
21	aa(c/g) tac (t/a)gg ^b^	Asn/Lys Tyr Trp/Arg	none
22	tg(c/g) (t/g)ac ccg ^b^	Cys/Trp Tyr/Asp Pro	none
23	deletion in randomized region ^c^	deletion in randomized region	frameshift
24	deletion in randomized region ^c^	deletion in randomized region	frameshift

^a^ In the sequenced region (positions 5800–5960 of the virus RNA encoding amino acids 121–174 of 3C). ^b^ Nucleotide heterogeneity in the randomized region. ^с^ Deletion in the randomized region, as judged by its unchanged flanking sequences.

**Table 4 viruses-11-00479-t004:** Specific infectivity of pools with different numbers of variants.

No.	RNA, µg	Number of Variants in the Pool	Day of Plaque Appearance and Their Size ^a^	Number of Plaques	Specific Infectivity, pfu/µg RNA ^b^
1	1.5	31	3, minute	77, 126	67.7
2	2	100	3, small	9, 20	7.25
3	1.5	102	5, small	102, 150	84
4	1.5	116	5, small	100, 100	66.7
5	1.5	148	5, minute	20, 39	19.7
6	1.5	185	5, minute	20, 50	23.4
7	2	250	3, minute	3, 6	2.25
8	1.5	271	3, minute	103, 130	77.7
9	1.5	297	3, minute	89, 90	59.7
10	2	350	3, small	17, 23	10
11	3	525	3, small	3, 18	3.5
12	1.5	630	3, minute	84, 130	71.4
13	1.5	666	3, small	34, 85	39.7
14	3	850	3, small	15, 30	7.5
15	3	~1750	3, large	5, 6	1.8
16	3	~1750	3, large	1, 2	0.5
17	2	~2250	3, large	14, 26	10
18	3	~2250	3, large/minute	16, 21	6.2
19	3	~2250	3, large/minute	43, 72	19.2

^a^ The cultures were observed for plaque appearance at days 3, 5, and 6 p.t. The day at which the plaques were first noticed and their size are indicated. The sizes of large, small, and minute plaques were >3 mm, 1–3 mm, and <1 mm, respectively. ^b^ The specific infectivity of the transcript corresponding to the full genome of wild type poliovirus, strain Mahoney, was 7.1 × 10^4^ pfu/µg.

**Table 5 viruses-11-00479-t005:** The prevalence (%) of different nucleotides in 22 unique sequences of 3C codons 154–156 of RNA of viable viruses.

Nucleotides	Codons of the 3C-Coding Region								
	154			155			156		
	N_1_	N_2_	N_3_	N_1_	N_2_	N_3_	N_1_	N_2_	N_3_
G	35	9	54	100	100	86	0	68	86
C	9	27	41	0	0	14	41	5	9
U	13	64	5	0	0	0	14	9	0
A	43	0	0	0	0	0	45	18	5

**Table 6 viruses-11-00479-t006:** Nucleotide and amino acid sequences in the randomized region of the selected viable viruses and their genetic stability ^a^.

No.	Primary Sequence ^b^		Sequence after 5–6 Additional Passages	
	Nucleotides	Amino Acids	Nucleotides	Amino Acids
1	**acg ggg aaa**	TGK	not done	not done
2	acg ggg cgg	TGR	acc ggg cgg	TGR
3	**acc ggg cgc**	TGR	not done	not done
4	acc ggg cgg	TGR	not done	not done
5	**gug ggg aag**	VGK	gug ggg aag	VGK
6	**guc ggg aag**	VGK	not done	not done
7	**guc ggg cgg**	VGR	guc ggg cgg	VGR
8	**gug ggg agg**	VGR	not done	not done
9	gug ggg cgg	VGR	not done	not done
10	**guu ggg agg**	VGR	not done	not done
11	**auc ggg agg**	IGR	auc ggg agg	IGR
12	ugc ggg aag	CGK	ugc ggg aag	CGK
13	aug gg ccgg	MGR	acg ggc cgg	TGR
14	uug ggc cgg	LGR	gug ggc cgg	VGR
15	cuc ggg agg	LGR	not done	not done
16	cug ggg cgg	LGR	not done	not done
17	(a/g)ug ggg cgg	(М/V)GR	gug ggg cgg	VGR
18	gug ggg aug	VGM	gug ggg aag	VGK
19	ugc ggg ugc	CGC	ugc ggg cgc	CGR
20	auc ggg ugg	IGW	auc ggg cgg	IGR
21	ugu acg ggc ucg ^c^	С_153_TGS	cgu acg ggc ucg	R_153_TGS
22	cgu acg ggg aug ^c^	R_153_TGM	(u/c)gu acg ggg a(u/a)g	(C/R)_153_TG(K/M)

^a^ The nucleotide sequences at positions 5897–5905 of the poliovirus genome and the corresponding amino acid sequences. In the case of changes after passages, the initially determined and changed nucleotides and amino acids are underlined. Parentheses indicate heterogeneous positions. ^b^ Initially determined sequences are given for the viruses isolated from the primary plaques (highlighted in bold) and for the viruses subjected to one or more passages to obtain sufficient material for the sequencing. ^c^ In addition to the changes in the randomized region, the codon at the preceding positions 5894–5896 (ugu) and the encoded amino acid (С_153_) were changed upon further passages of the primary isolate 21 and were already changed (and underwent further alterations) in the primary isolate 22.

**Table 7 viruses-11-00479-t007:** Phenotype and genetic stability of constructed genomes.

No.	Codons (153)154–156 of 3C	Amino acid (153)154–156 of 3C	Day of Plaque Appearance	Plaque Size	Relative Specific Infectivity ^a^	Changes in Recovered Viruses ^b^	
						Nucleotide	Amino Acids
1	acu ggg aaa	TGK	3	large	1	no changes	
2	guc ggg aag	VGK	3	large	0.8	no changes	
3	auu ggc aaa	IGK	3	large	1.2	no changes	
4	ugc ggg aag	CGK	3	large	1.4	no changes	
5	acg gg gcgg	TGR	3	large	1.6	not done	
6	aug ggg aaa	MGK	3	minute	0.6	a_5897_g	M_154_V ^c^
						a_5897_u	M_154_L
						a_5897_(a/g) u_5898_(u/c)	M_154_(V/T)
						c_5699_u	P_88_S
7	uug ggg aaa	LGK	6	small/heterogeneous	0.2	u_5897_g	L_154_V
						c_5700_u	P_88_L
8	ucg ggg aaa	SGK	8	solitary plaques ^d^	0.0007	c_5898_g g_5899_c	S_154_C
9	acg ggg aug	TGM	7–8	solitary plaques ^d^	0.07	u_5896_c	C_153_R ^c^
						u_5904_a	M_156_K
10	acg ggc ucg	TGS	7–8	solitary plaques ^d^	0.003	c_5700_u	P_88_L
						c_5699_a	P_88_T
						g_5738_a	V_101_I
						g_6281_a	D_282_N
11	(cgu) acg ggg aug	(R)TGM	3–5	small/heterogeneous	0.3	g_5921_a	V_162_I^c^
						a_5846_g	M_137_V
						a_5775_u	Y_113_F
						a_5625_u	E_63_V
						g_5777_a	V_114_I
12	(cgu) acg ggc ucg	(R)TGS	3–5	small/heterogeneous	0.02	c_5699U_	P_88_S^c^
						g_5798_c	E_121_Q
						g_5738_a	V_101_I
						a_5888_c	I_151_L
13	acu ggg gaa	TGE	no plaques				
14	gug ggg gaa	VGE	no plaques				
15	aug ggc cgg	MGR	4–6	small	1.3	a_5897_g	M_154_V^c^
						u_5898_c	M_154_T
						u_5898_(u/c)	M_154_(T/M)
16	auc ggg ugg	IGW	6–8	small/heterogeneous	0.2	u_5903_c	W_156_R^c^
17	ugc ggg ugc	CGC	7–8	solitary plaques ^d^	0.0015	u_5903_c	C_156_R
18	gug ggg aug	VGM	7–9	solitary plaques ^d^	0.0015	u_5904_(u/a)	M_156_(K/M)
19	(cgu) acg ggc ucg	(R)TGS ^e^	3	small	1.2	not done	
20	(cgu) acg ggc ucg	(R)TGS ^f^	3	small	1.4	not done	

^a^ Specific infectivity of the RNA-transcript of poliovirus strain Mahoney determined in the respective experiments, where (5.25–6.5) × 10^2^ pfu/µg, is assumed to be 1.0. ^b^ Positions 5500–6500 of the poliovirus RNA were sequenced. ^c^ Such changes were found in several clones. ^d^ Late solitary plaques after transfection with a high dose (~1 μg) of the transcript. ^e^ Containing the engineered P_88_S mutation. ^f^ Containing the engineered I_151_L mutation.

**Table 8 viruses-11-00479-t008:** Partial nucleotide and amino acid sequences and genetic stability of viable viruses selected from the genomes with both randomized apical octanucleotide of domain *d* and nonanucleotide encoding the TGK motif of 3C.

Pool No. ^a^	Number of Isolates	Primary Sequence				Sequence after 2–6 Additional Passages				
		nt. 61–68 of *ori*L	Tetraloop Consensus	Codons 154–156 of 3C		Passage	Nt 61–68 of *ori*L	Tetraloop Consensus	Codons 154–156 of 3C	
				Nucleotides	Amino acids				Nucleotides	Amino Acids
1	1	ugAU(U/G)Uca + insertion G_14_	no consensus	gug ggg agg	VGR	5	ugGUN(U/A)ca + insertion G_14_	no consensus/GNUA	gug ggg agg	VGR
						6	ugGUUAca + insertion G_14_	GNUA	gug ggg agg	VGR
2	1	uaGCUCua	no consensus	no data		3	uaGCUCua	no consensus	cug ggg agg	LGR
						5	uaGCU(C/A)ua	GNUA	gug ggg agg	VGR
3	1	uaUCAGug	YNMG	acg ggg cgc	TGR	2	uaUCAGug	YNMG	acg ggg cgc	TGR
						5	uaUCAGu(g/a)		acg ggg cgc	TGR
	4 ^b^					2	uaUCAGug	YNMG	acg ggg cgc	TGR
						5	uaUCAGug		acg ggg cgc	TGR
4	1	auUUAUgu + insertion C_119_	no consensus	gug ggg agg	VGR	2	guUUAUgc + insertion C_119_	no consensus	gug ggg agg	VGR
						3	guUUA(U/G)gc + insertion C_119_	YNMG	gug ggg agg	VGR
						5	guUUAGgc + insertion C_119_		gug ggg agg	VGR
	1					3	auUUAGgu + insertion C_119_	YNMG	gug ggg agg	VGR
	3 ^b^					3	auUUAUgu + insertion C_119_	no consensus	gug ggg agg	VGR

^a^ Pools contained transcripts from 1000–15,000 plasmid clones. ^b^ All isolates demonstrated the same genetic alterations upon passages.

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
