# Peer review of "Characterization of Mutational Tolerance of a Viral RNA–Protein Interaction"

_viruses, 2019, doi:10.3390/v11050479_

Round 1
Reviewer 1 Report
Prostova and colleagues have submitted a manuscript that examines the mutational space and tolerance of a well-characterized and essential RNA-protein interaction in poliovirus. In particular they examined a tri-amino acid motif (TGK) within the 3CD protein that interacts with the d domain of the oriL (or cloverleaf structure) in the 5’ untranslated region of the poliovirus genome. The authors applied randomized mutagenesis of this particular region and then examined the effect on specific infectivity, plaque assay, replication and RNA-protein interactions. This is an extensive study and the manuscript is well presented and written. Specific comments are noted below:
1. The creation of mutant genomes and initial set-up is quite elaborate. It would help the reader if the authors included a schematic/flow chart of the work flow.
2. To demonstrate reproducibility, it is important that the authors note in each figure legend, the number of times the experiments were undertaken. For example: in Figure 4, the authors note that “Each panel presents the results of a single experiment.” Was the experiment only undertaken once or is it representative of x number of experiments?
3. Figure 5: It would be more informative if the authors showed a bar graph of the binding interaction between 3CD and oriL and include standard deviations and statistical analyses.
4. Figure 5: The authors show that the CGK, SGK, MGR, TGM and RTGM marginally bind the oriL. One possible explanation for poor binding is that the mutation changes the conformation of the protein rendering it nonfunctional, or even degraded. The authors should include a western blot showing expression of 3CD wild-type and mutant proteins.
5. Prostova uncover a number of mutations of the tri-amino acid motif that affect plaque size (specific infectivity), replication and RNA-3CD interactions. Moreover on page 19 they discuss possible mechanisms and effects on the protein structure. The authors might consider including a figure of the structure and then show TGK and other amino acids surrounding the motif. This would provide a visual insight into how these mutations changes might/might not impact function.
6. While the goal of this manuscript was to examine mutational tolerance, have some of these mutations been identified in circulating strains? It would be worthwhile for the authors to include a paragraph in the discussion addressing this.
7. Minor comments:
a. Line 122: “To generate plasmids with randomized both the …” something is missing in the first part of the sentence.
b. Line 199: typo -acida > should be acids
Author Response
1. As requested, a schematic chart is added – Figure 2.
2. Clarification about the reproducibility of the experiments shown in Fig. 6 is made in its legend.
3. The bar graph to former Fig. 5 (now #7) is added.
4. To demonstrate the quality of the engineered mutant 3CD proteins a new Fig. 3 is added.
5. As requested, a new Fig. (#9) illustrating some features of the crystal structure of 3C is added. The relevant methodical procedures are added to Materials and Methods (l.l. 211-213).
6. Responding to the comment of the reviewer a specification concerning the conservation of amino acids in the TGK motif is made (l.l. 518-519).
7ab. The suggested minor corrections are made.
Reviewer 2 Report
In this manuscript, Prostova and co-authors study experimentally mutational tolerance and
interactions between TGK domain of 3C protein and the so called clover leaf structure in the 5' end of the poliovirus genome.
The rationale behind the current experiments is derived from the error-prone replication of single stranded RNA viruses, which is likely to accumulate fitness-decreasing mutations in the virus genome. The viruses are therefore expected to show increased mutational tolerance retaining the virus phenotype. Experimental approches, such as the ones used in this manuscript, to study virus evolution are highly warranted in order to increase the knowledge of evolutionary mechanisms used by RNA viruses, since due to the extermely rapid replication during the acute virus infections, detailed step-by-step processes and compensational mutations are often difficult to observe in nature.
The authors first by randomized the sequence of TGK coding region followed by in vitro selection experiments. The fitness-affecting mutations are further validated using reverse genetic approach. In addition, the phenotypes of distinct sets of mutations are assessed by studying replication efficiency and the interaction between the clover-leaf and 3CD. The authors show that many of the non-lethal mutations have seemingly no effect on the virus fitness (in vitro), pin-point the details of accepted mutations and show that many of the slightly deletorious mutations are followed by compensating mutations.
The manuscript is well written, the experiments are characterized in detail, and the results are well discussed (including the limitations of the study).
I have only few minor issues on this manuscript:
The electrophoric mobility assay should be briefly characterized in the methods section.
The tables would be more readable if upper case abbreviation for nucleotides and single letter abbreviations (instead of three letter) were used for amino acids in all tables (see table 3).
Author Response
1. A sentence explaining the EMSA assay is added to Materials and Methods (l.l. 207-209).
2. The reviewer suggested using upper case abbreviation for nucleotides and single letter abbreviations (instead of three letter) for amino acids. We feel however that in this case identical upper case letters would have different senses in the neighboring columns. No changes were made.